# Dynamic Perfluorinated Gas MRI Shows Improved Lung Ventilation in People with Cystic Fibrosis after Elexacaftor/Tezacaftor/Ivacaftor: An Observational Study

**DOI:** 10.3390/jcm11206160

**Published:** 2022-10-19

**Authors:** Jennifer L. Goralski, Sang Hun Chung, Agathe S. Ceppe, Margret Z. Powell, Muthu Sakthivel, Brian D. Handly, Yueh Z. Lee, Scott H. Donaldson

**Affiliations:** 1Division of Pulmonary and Critical Care Medicine, University of North Carolina at Chapel Hill, Chapel Hill, NC 27599, USA; 2Marsico Lung Institute/UNC Cystic Fibrosis Center, University of North Carolina at Chapel Hill, Chapel Hill, NC 27599, USA; 3Division of Pediatric Pulmonology, University of North Carolina at Chapel Hill, Chapel Hill, NC 27599, USA; 4Department of Biomedical Engineering, University of North Carolina at Chapel Hill—North Carolina State University, Chapel Hill, NC 27599, USA; 5Department of Radiology, University of North Carolina at Chapel Hill, Chapel Hill, NC 27599, USA; 6Biomedical Research Imaging Center, University of North Carolina at Chapel Hill, Chapel Hill, NC 27599, USA

**Keywords:** cystic fibrosis, MRI, contrast gas, ventilation, structure, perfluoropropane

## Abstract

The availability of highly effective CFTR modulators is revolutionizing the treatment of cystic fibrosis (CF) and drastically improving outcomes. MRI-based imaging modalities are now emerging as highly sensitive endpoints, particularly in the setting of mild lung disease. Adult CF patients were recruited from a single center prior to starting treatment with E/T/I. The following studies were obtained before and after one month on treatment: spirometry, multiple breath nitrogen washout (MBW), ^1^H UTE MRI (structural images) and ^19^F MRI (ventilation images). Changes between visits were calculated, as were correlations between FEV_1_, lung clearance index (LCI), MRI structural scores, and MRI-based ventilation descriptors. Eight subjects had complete datasets for evaluation. Consistent with prior clinical trials, FEV_1_ and LCI improved after 28 days of E/T/I use. ^1^H UTE MRI detected improvements in bronchiectasis/airway wall thickening score and mucus plugging score after 28 days of therapy. ^19^F MRI demonstrated improvements in fractional lung volume with slow gas washout time (FLV↑tau2) and ventilation defect percentage (VDP). Improvements in FLV↑tau2 and VDP correlated with improvement in FEV_1_ (r = 0.81 and 0.86, respectively, *p* < 0.05). This observational study establishes the ability of ^19^F MRI and ^1^H UTE MRI to detect improvements in lung structure and function after E/T/I treatment. This study supports further development of ^19^F MRI and ^1^H UTE MRI as outcome measures for cystic fibrosis research and drug development.

## 1. Introduction

Recent advances in cystic fibrosis (CF) research and clinical care have pushed the community into welcome, although unfamiliar, territory. In 2019, the United States Food and Drug Administration granted approval of elexacaftor/tezacaftor/ivacaftor for people with cystic fibrosis, 12 years or older, carrying at least one copy of the F508del mutation. Pivotal phase three studies showed improvement in quality of life, lung function, and sweat chloride values [1,2]. In 2020, the approval indication was expanded to include 177 additional mutations, based on in vitro theratyping. In 2021, safety and efficacy were demonstrated in children with CF between the ages of 6 to 11 years [3], which led to additional label expansion. With ongoing studies in children aged 2–5 years (NCT04537793), the future of CF care likely will reflect more modest lung disease in aging adolescents and adults.

Historically, clinical trials in CF have used pulmonary exacerbation frequency, patient reported outcomes, and spirometry as validated or surrogate endpoints for clinical trials that target CF lung disease. However, it is well known that these outcome measures have limited value in mild lung disease [4]. Recently, the lung clearance index (LCI), derived from multiple breath washout (MBW) tests, and quantitation of ventilation defect percentages (VDP) from hyperpolarized gas MRI have been explored as more sensitive outcome measures. Given the trend toward milder lung disease in the era of highly effective CFTR modulator therapy, the need for novel, more sensitive outcome measures is growing [4,5,6].

Validation of new outcome measures requires that we establish biologic plausibility and clinical relevance, understand short term and long term test variability, and characterize the responsiveness of the test to treatment or other changes in disease state [7]. This process relies upon comparisons to other measures of disease severity and requires an assessment of test acceptability to study participants. Our group has been studying dynamic ventilation imaging with fluorine-enhanced MRI (^19^F MRI) as a potential biomarker for the study of mild CF lung disease. ^19^F MRI characterizes the regional kinetics of a tracer gas (perfluoropropane) wash-in and wash-out over serial breaths. This contrasts with hyperpolarized gas MRI studies, which typically identify non-ventilated lung regions after a single breath of tracer gas. Although functional MRI-based techniques have proven to sensitively detect lung disease, characterization of lung structure is an equally important and relevant goal. The gold standard for characterizing CF lung disease is high resolution CT scans. An alternative approach that has been shown to correlate with HRCT, does not require radiation, and can be performed in the same sitting as functional MRI scans, is ultrashort echo time MRI (^1^H UTE) [8]. The recent approval of elexacaftor/tezacaftor/ivacaftor for people with CF provided a unique opportunity to test the ability of ^1^H UTE and ^19^F MRI to spatially characterize changes in lung structure and ventilation kinetics, respectively, in response to this highly effective therapy. The study was designed to also assess the correlation between MRI-based descriptors of lung disease and established measures of lung function (FEV_1_ and LCI).

## 2. Materials and Methods

This study was approved by the Institutional Review Board at the University of North Carolina at Chapel Hill and all subjects provided signed informed consent. Study participants were recruited prior to commencing treatment with elexacaftor/tezacaftor/ivacaftor and completed identical study procedures at baseline (before starting elexacaftor/tezacaftor/ivacaftor) and at the second study visit (1 month after starting treatment). Key inclusion criteria included age ≥ 18 years, diagnosis of cystic fibrosis by standard genotypic/phenotypic criteria, forced expiratory volume in 1 s (FEV_1_) ≥ 40% predicted, and intent to start treatment with elexacaftor/tezacaftor/ivacaftor. Subjects with a smoking history, contraindication to MRI, or unstable disease were excluded from the study. All other prescribed medications were allowed during the observational study, but participants were asked not to make changes to their medication or airway clearance regimen other than elexacaftor/tezacaftor/ivacaftor initiation.

After informed consent, subjects performed multiple breath nitrogen washout (MBW; Exhalyzer^®^ D, EcoMedics, Spiroware 3.1.6, Dürnten, Switzerland) per standard protocol [9]. Spirometry was then performed using a KoKo spirometer (Nspire Health, Longmont, CO, USA) according to American Thoracic Society standards [10] using the Hankinson (NHANES III) [11] reference equations. Next, ^1^H UTE MRI was performed to characterize lung structure. Finally, ^19^F MRI during wash-in and wash-out of inhaled perfluoropropane was performed to characterize lung ventilation, as previously described [12]. Briefly, ^1^H UTE MRI was performed with a 3T MRI scanner (PRISMA, Siemens Medical Systems, Malvern, PA, USA) using the following parameters: fast low angle shot three-dimensional imaging (FL3D); echo time (TE), 0.05 ms; repetition time (TR), 2.42 ms; flip angle (FA), 5; resolution, 2.14 mm × 2.14 mm; slice thickness, 2.5 mm; number of slices, 103. Scans were acquired in the supine position with the arms positioned above the head, at full inspiration and expiration, using an embedded body coil. Then, using a commercially available 8-channel ^19^F-tuned chest coil (ScanMed, LLC, Omaha, NE, USA), images were obtained while the participant breathed a pre-mixed medical grade gas mixture (79% perfluoropropane/21% oxygen) via a non-rebreathing circuit under IND 122,215. During gas wash-in, a technician guided the participant’s breathing pattern as follows: a tidal breath of perfluoropropane (inhalation and exhalation) followed by a full inspiration and 15-s breath hold, during which images were obtained, and then exhalation. This breathing sequence was repeated 5 times before switching the gas mixture to room air to initiate the gas wash-out phase. An identical breathing pattern was used during gas wash-out, until no visible ^19^F signal was visualized in lung images monitored in real time. MR parameters used for the ^19^F images were as follows: 3D Volumetric interpolated breath-hold examination (VIBE); TR: 13 ms; TE: 1.61 ms; averages, 2; FA: 70; resolution, 6.25 mm × 6.25 mm; slice thickness, 15 mm; number of slices, 18. Pulse oximetry, heart rate, and exhaled CO_2_ were monitored throughout the duration of the scan as safety measures.

Two thoracic radiologists scored ^1^H UTE MRI images blindly and independently using the Eichinger criteria [13], where disease burden per lobe (treating lingula as an independent lobe) is rated as 0 (no abnormality), 1 (<50% of lobe involved), or 2 (>50% of lobe involved). Morphologic assessments include (i) bronchial wall thickening and/or bronchiectasis; (ii) mucus plugging; (iii) sacculation or abscess; (iv) consolidations; and (v) pleural reaction including effusion or pneumothorax. Because we did not use intravenous contrast in these studies, perfusion was not assessed by the reviewers and the total maximum score for any patient was 60, with higher scores indicating higher disease burden.

For ^19^F MRI images, ^19^F signal intensity was plotted over time in each voxel within the lung region, using the boundaries manually defined by both the anatomic ^1^H scans and the matched ^19^F images. The percentage of lung with ventilation defects (VDP) after the 5th inspiratory cycle was measured, using the 95th percentile of background noise on the last wash-in scan as the threshold value defining absence of ventilation. In ventilated lung regions, a bi-exponential model was used to fit ^19^F signal over time curves in each voxel to estimate the rate constant that characterizes the kinetics of gas wash-out (tau2). The threshold used to define an abnormally slow tau2 value (>163 s) was derived from a prior study that included healthy and CF participants [12]. The fraction of the total lung volume with slow gas wash-out kinetics (FLV↑tau2) was then calculated for lung regions without an overlapping full ventilation defect. A combined abnormal ventilation score, consisting of the summation of VDP and FLV↑tau2 percent, was calculated for each image series.

Statistical analysis was performed using JMP Pro 14 (SAS Institute, Cary, NC, USA). Reader agreement on structural MRI scores was assessed by calculating the inter-observer correlation coefficient. Comparisons between baseline and post-elexacaftor/tezacaftor/ivacaftor data were made with paired Student’s *t*-tests for continuous data and chi-square test for categorical data. A *p*-value of < 0.05 was considered significant. The relationship between each imaging and clinical parameter (FVC, FEV_1_, FEF25:75, LCI, VDP, FLV↑tau2, abnormal ventilation score, overall morphology score, bronchiectasis/airway wall thickening sub-score, and mucus plugging sub-score) were represented via the Spearman correlation coefficient.

## 3. Results

Eleven subjects were recruited and eight completed both visits (Figure 1). Early termination of 3 subjects occurred due to: (1) anxiety in the MRI scanner that precluded image acquisition at baseline; (2) FEV_1_ below the clinical baseline at enrollment; and (3) a CF exacerbation occurring between visits that excluded the subject from the post elexacaftor/tezacaftor/ivacaftor visit. No adverse events related to perfluoropropane gas inhalation were noted among the participants in this study. Median time between the baseline visit and start of elexacaftor/tezacaftor/ivacaftor was 5.5 days (range 1–22).

Table 1 and Figure 2A,B show demographic and lung function data before and after treatment. In these adult participants, lung function was mild to moderate in severity at baseline and improved significantly after 4 weeks of elexacaftor/tezacaftor/ivacaftor treatment. Similarly, the homogeneity of ventilation, as assessed with the LCI, also improved significantly. The magnitude of these improvements is consistent with the data seen in the phase 3 clinical trials for elexacaftor/tezacaftor/ivacaftor [1,2].

### 3.1. Pre–Post Elexacaftor/Tezacaftor/Ivacaftor Lung Morphology Comparison

Table 2 and Figure 2C presents lung structure data obtained at baseline and after 1 month of treatment. Of note, there was excellent agreement between reader scores (r = 0.82, *p* < 0.001); therefore, mean reader scores are presented. The overall morphologic structure score improved significantly, driven primarily by changes in the bronchiectasis/airway wall thickening and mucus plugging sub-scores. Other score components (e.g., abscess/sacculation, consolidation) were uncommon features at baseline and therefore did not change significantly. The presence of bronchiectasis/airway wall thickening decreased from 65% to 50% of lobes assessed after treatment. Similarly, the presence of mucus plugging decreased from 58% to 29% of lobes affected after elexacaftor/tezacaftor/ivacaftor.

At baseline, the right upper lobe and left upper lobe were the most severely affected lobes based on the lobar morphology structure score (Table 3 and Figure 3), and this remained true at the follow-up visit. However, structural scores improved with elexacaftor/tezacaftor/ivacaftor treatment in all lobes, with the exception of the lingula, which showed modest worsening in this small sample.

### 3.2. Pre–Post Elexacaftor/Tezacaftor/Ivacaftor Ventilation Comparison

We observed a statistically significant improvement in ^19^F MRI ventilation parameters after elexacaftor/tezacaftor/ivacaftor treatment (Table 4 and Figure 2D). Both the fraction of lung with no ventilation (i.e., VDP) and the fraction of lung with slow gas washout (i.e., FLV with prolonged tau2 value) improved with treatment. Review of paired ^19^F MRI images (Figure 4) suggest that some regions characterized as total ventilation defects at baseline converted to a slow ventilating region, indicating that these defects are potentially reversible.

### 3.3. Correlation of Results

We next examined the relationship between imaging endpoints and traditional lung function measurements, i.e., FEV_1_ and LCI. Strong correlations between FEV_1_ and MRI ventilation (VDP, FLV↑tau2, abnormal ventilation score) and structure scores (overall morphology score, bronchiectasis and mucus plugging sub-scores) were noted at baseline (Table 5 and Figure 5). At follow-up, similar strong correlations with FEV_1_ were observed with the exception of the mucus plugging sub-score. LCI also correlated strongly with total morphology, bronchiectasis, and mucus plugging scores, and correlated moderately well with FLV↑tau2 (*p* = 0.08 and <0.05 at baseline and follow-up, respectively). However, LCI did not significantly correlate with VDP at either visit (r < 0.6; *p* = NS). A correlation trend between the changes in FEV_1_ and FLV↑tau2 was noted (r = −0.575, *p* = NS), but other correlations between the calculated changes in other imaging and lung function endpoints were generally weak. Finally, we observed a strong, significant correlation between ventilation (combined abnormal ventilation score) and structure scores (Figure 5) both at baseline (r = 0.89, *p* = 0.006) and at follow-up (r = 0.78, *p* = 0.03).

## 4. Discussion

This study capitalized on the approval of a new, highly effective CFTR modulator. In this observational study, we examined treatment responses using ^19^F MRI and ^1^H UTE MRI and correlated imaging and traditional lung function changes after 28 days on elexacaftor/tezacaftor/ivacaftor.

Participants in this study were adults with relatively preserved lung function (median percent predicted FEV_1_ of 84.5%, with a range of 64–105% predicted). Consistent with previously reported clinical trial findings, we observed a significant improvement in FEV_1_ after 28 days of treatment with elexacaftor/tezacaftor/ivacaftor. We also found a reduction in median LCI score and LCI z-score, which indicates improvement in ventilation homogeneity and reflects reduced obstruction of small airways. Despite participants having relatively mild structural abnormalities at baseline compared to prior studies [14,15], we were able to detect significant improvements in airway wall abnormalities and mucus plugging with treatment.

The improvements in mucus plugging seen on MRI are consistent with the mechanisms of action of elexacaftor/tezacaftor/ivacaftor, as highly effective CFTR modulators have been shown to enhance mucociliary clearance [16]. We also noted a reduction in the number of lobes that displayed any amount of mucus plugging or bronchiectasis, indicating that not only was the lobar severity of these sub-scores improved, but after elexacaftor/tezacaftor/ivacaftor, some lobes were now characterized as structurally normal. As expected in CF, upper lobe disease was the predominant finding, but there was a strong tendency for improvement in all lobes.

While morphologic findings on ^1^H MRI can lead to an overall estimate of the burden of structural disease, these images alone do not offer a functional assessment of the lung. Therefore, we characterized the dynamic wash-out kinetics of the ^19^F signal provided by PFP gas before and after treatment with elexacaftor/tezacaftor/ivacaftor. We have previously shown that ^19^F MRI is adequately sensitive to differentiate CF patients with FEV_1_ ≥ 80% from healthy controls [12]. The data provided here indicate that this technique is also is able to detect the substantial treatment effect provided by elexacaftor/tezacaftor/ivacaftor. Simultaneous use of two MRI modalities furthers our understanding of the relationship between improvements in lung structure and ventilation, as well as their relationship to traditional lung physiologic measurements (i.e., FEV_1_ and LCI).

Dynamic characterization of ventilation with ^19^F MRI suggests that ventilation abnormalities exist on a continuum [17], a distinction that may be under-appreciated with single breath hyperpolarized gas studies. In fact, slowly ventilating regions identified with a multi-breath gas tracer protocol may be misclassified either as a full ventilation defect or as “normal” when utilizing a single breath of hyperpolarized gas protocol, depending upon the voxel signal after a single breath and the threshold value used to define a ventilation defect [17]. Slowly ventilating regions may still contribute to gas exchange and, therefore, may impact lung physiology differently than a true ventilation defect. With ^19^F MRI, we are able to distinguish between non-ventilating and poorly ventilating lung regions, report the fraction of lung in either category and observe the transformation of voxels from non-ventilating to ventilating after treatment with elexacaftor/tezacaftor/ivacaftor. As shown in Table 4, we observed an overall reduction in abnormally ventilating lung regions (i.e., VDP plus FLV↑tau2) by 4.37%, representing a 33% relative reduction from baseline to 28 days after starting E/T/I. While both component parts of the combined score independently improved significantly, it was noted that some voxels characterized as VDP at baseline shifted to the slow emptying category (FLV↑tau2) at follow-up, thus minimizing the overall change in FLV↑tau2. This transition from “absent” ventilation to “poor” ventilation nonetheless likely represents an improvement in lung ventilation.

When examining the relationship between different endpoints, significant correlations were found between FEV1 and the other measures studied, including LCI, morphologic structure score, bronchiectasis/airway wall thickening sub-score, mucus plugging sub-score, VDP, and FLV↑tau2 at each visit, suggesting that these variables are closely related. In contrast, the change in FEV1 between visits poorly correlated with the change in these same variables. A treatment effect captured by these endpoints may, therefore, be reflecting different aspects of lung physiology. Importantly, we also showed a strong correlation between the extent of structural disease characterized by ^1^H MRI with the extent of ventilation abnormalities, as demonstrated by both the VDP and FLV↑tau2. While perhaps not surprising, this helps to validate that observed ventilation abnormalities are truly a function of structural airways disease and not likely reflecting transient bronchoconstriction or other imaging artifacts. Our findings are also consistent with recent publications [14,18] that showed lung structural improvements after treatment with elexacaftor/tezacaftor/ivacaftor are driven by changes in bronchiectasis/airway wall thickening and mucus plugging. Our study adds to the growing body of literature that structural improvements are detectable within 4 weeks of starting highly effective CFTR modulator therapy and correspond to improvements in ventilation defects detected by ^19^F MRI.

Our study has several limitations. First, the small sample size reflects the rapid clinical uptake of elexacaftor/tezacaftor/ivacaftor post-approval, which limited the opportunity to enroll patients prior to starting treatment. More extensive studies will be needed to be completed before such techniques could be routinely used as outcome measures for clinical trials. Next, the day-to-day variability of VDP and FLV↑tau2, as obtained by ^19^F MRI, has not yet been established. A separate study to evaluate the reproducibility of this technique is currently underway.

Another potential limitation relates to the enrollment of adults with mild disease. It is uncertain, therefore, whether our findings will translate to pediatric populations, or those with more significant lung disease. Of note, participants in our study generally had milder structural disease scores than those in other published studies [14,15] although we should also consider whether differences in scoring systems contributed to a milder overall score. The most robust magnetic resonance imaging scoring systems use a combination of morphologic score and perfusion score, requiring administration of intravenous contrast [13,19,20]. These studies show a high prevalence of perfusion abnormalities (~80%) in people with CF. Modified scoring systems may use a more sensitive ^1^H UTE technique to permit scoring of bronchiectasis and airway wall thickening separately [15,21], while eliminating the perfusion component.

Finally, we used an embedded MRI proton coil rather than a high-sensitivity chest coil during ^1^H imaging to avoid patient repositioning during coil exchanges that would have impacted image registration between techniques. This limited our ability to distinguish true bronchiectasis from airway wall thickening, particularly in peripheral lung zones. Use of an optimized receiver coil in ^1^H UTE MRI studies would improve spatial resolution substantially [15,22] and allow for further characterization of true bronchiectasis and airway wall thickening. Development of a dual-tuned ^1^H/^19^F coil would also avoid participant repositioning and image misalignment during combined structure/ventilation imaging studies, but such a coil is not currently available.

Therefore, our overall MR scores may underrepresent the magnitude of structural disease that might be reported in a similar population using a more sensitive technique. Despite this shortcoming, our ability to detect structural changes in a small cohort of CF patients with a potentially less sensitive technique is noteworthy. Importantly, this study adds to a growing body of data that airway structural changes may, in fact, be reversible when a patient is treated with highly effective CFTR modulator therapy.

## 5. Conclusions

This real-world observational study of elexacaftor/tezacaftor/ivacaftor in people with CF demonstrated improvements in lung structure (via ^1^H UTE) and lung function (via ^19^F MRI) in response to this highly effective therapy. Strong correlations were found between MRI-based descriptors of lung disease (including VDP, FLV↑tau2, and morphology score) and established measures of lung function (FEV_1_ and LCI).

These data demonstrate that ^1^H UTE and ^19^F MRI are responsive to a treatment that improves the disease state in CF. Further studies are needed to evaluate the repeatability of these measures, and to assess the sensitivity of these endpoints to treatment effects that are less pronounced than those observed with elexacaftor/tezacaftor/ivacaftor. 

## Figures and Tables

**Figure 1 jcm-11-06160-f001:**
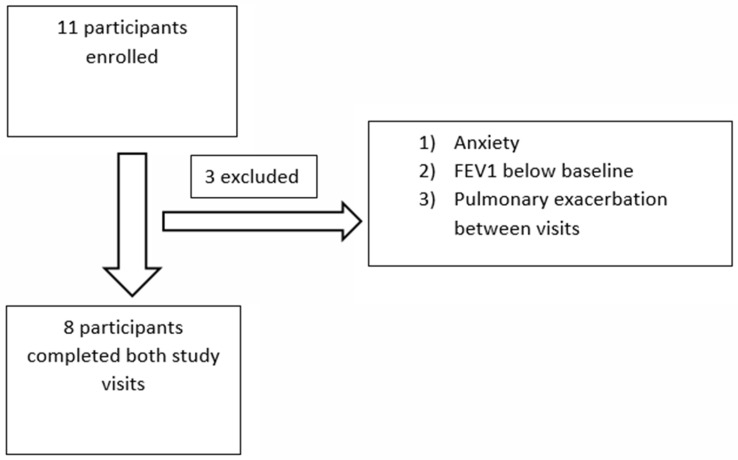
Flow diagram of participants enrolled in the observational trial. FEV_1_: forced expiratory volume in 1 s.

**Figure 2 jcm-11-06160-f002:**
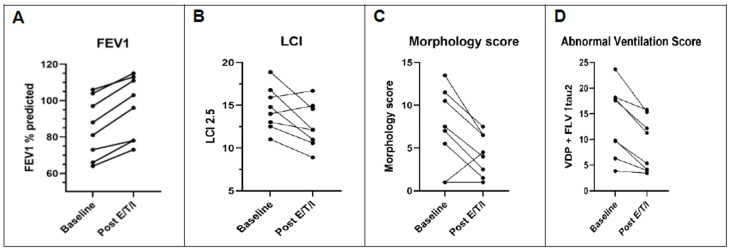
Effect of E/T/I on lung function, lung MRI morphology score, and lung MRI ventilation score in subjects with CF. Patient level changes from baseline to one month after elexacaftor/tezacaftor/ivacaftor. (**A**) FEV1. (**B**) LCI. (**C**) Morphology score from ^1^H MRI. (**D**) Abnormal ventilation score from ^19^F MRI, a summation of VDP and FLV↑tau2 values. E/T/I: elexacaftor/tezacaftor/ivacaftor. FEV_1_: forced expiratory volume in 1 s. LCI: lung clearance index. VDP: ventilation defect percentage. FLV↑tau2: fractional lung volume with prolonged gas washout time.

**Figure 3 jcm-11-06160-f003:**
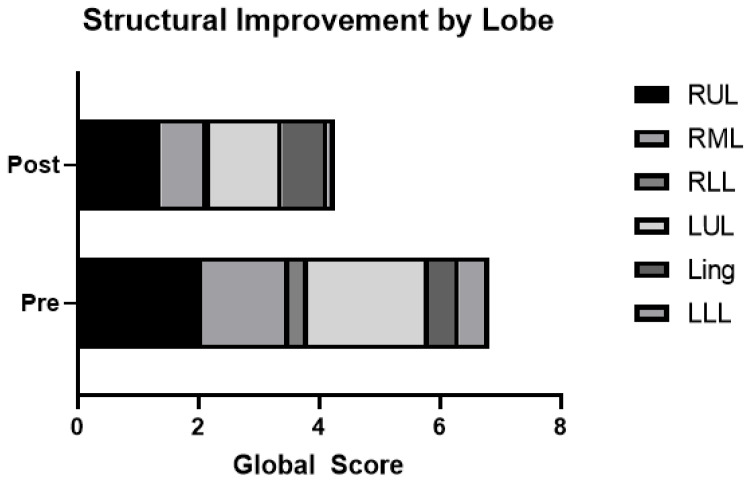
Morphology scores by lobe before and after elexacaftor/tezacaftor/ivacaftor. Structural improvement by lobe of lung, with higher scores representing worse disease. Ling: lingula; LLL: left lower lobe; LUL: left upper lobe; RML: right middle lobe; RLL: right lower lobe; RUL: right upper lobe.

**Figure 4 jcm-11-06160-f004:**
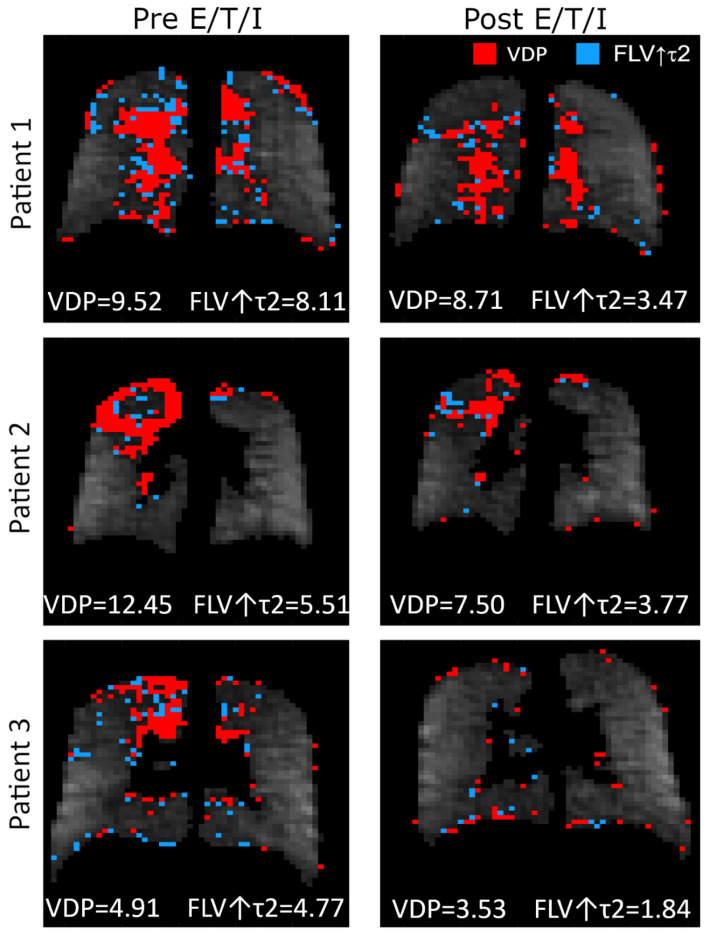
Effect of elexacaftor/tezacaftor/ivacaftor on the distribution of non-ventilating lung units. E/T/I: elexacaftor/tezacaftor/ivacaftor. VDP: Ventilation defect percentage. FLV↑τ2: fractional lung volume with prolonged gas washout time.

**Figure 5 jcm-11-06160-f005:**
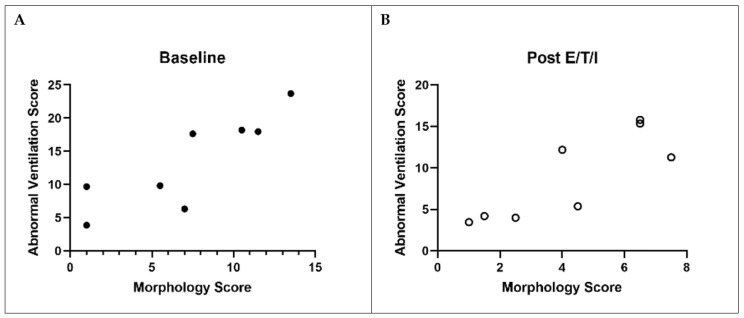
Correlation between lung structure and function before ((**A**), closed circles) and after ((**B**), open circles) elexacaftor/tezacaftor/ivacaftor. Correlation between combined abnormal ventilation score with overall morphology score at baseline and post elexacaftor/tezacaftor/ivacaftor. Spearman correlation at baseline: r = 0.89; *p* = 0.006. Spearman correlation at post elexacaftor/tezacaftor/ivacaftor: r = 0.78; *p* = 0.03. E/T/I: elexacaftor/tezacaftor/ivacaftor.

**Table 1 jcm-11-06160-t001:** Demographic and lung function data.

Total *N* = 8 with Paired Data	Baseline	Post Elexacaftor /Tezacaftor/Ivacaftor	Mean between Visit Change	*p*-Value
Female (*n*, %)	5 (62.5%)			
Age, years (median, range)	32 (22–45)			
FVC, liters (mean, SD)	4.36 (1.33)	4.67 (1.36)	0.31 (0.11)	<0.001
FVC, % predicted (mean, SD)	99.5% (11.46)	106.6% (9.97)	7.13% (1.96)	<0.001
FEV1, liters (mean, SD)	3.12 (1.12)	3.53 (1.27)	0.41 (0.20)	<0.001
FEV1 % predicted (mean, SD)	84.88% (16.55)	95.89% (17.32)	11.0 (0.78)	<0.001
FEF25:75, liters (mean, SD)	2.38 (1.32)	3.06 (1.72)	0.68 (0.40)	0.007
FEF25:75, % predicted (mean, SD)	60.38% (29.12)	77% (35.74)	16.63% (11.07)	0.004
LCI (mean, SD)	14.62 (2.53)	12.62 (2.59)	−2.00 (2.18)	0.036

FEF25:75: forced expiratory flow at 25:75%. FEV1: forced expiratory volume in 1 s. FVC: forced vital capacity. LCI: lung clearance index at 2.5% stopping point. SD: standard deviation.

**Table 2 jcm-11-06160-t002:** Structural MRI scores.

Total *N* = 8 with Paired Data	Baseline (Mean, SD)	Post E/T/I (Mean, SD)	Mean between Visit Change	*p*-Value
Morphology Structure score	7.19 (4.61)	4.25 (2.45)	−2.94 (3.22)	0.04
Bronchiectasis/airway wall thickening sub-score	3.81 (0.62)	2.63 (0.57)	−1.19 (0.60)	0.002
Mucus plugging sub-score	3 (0.5)	1.1875 (0.34)	−1.82 (0.44)	<0.001
Abscess/sacculation sub-score	0	0	0	--
Consolidation sub-score	0.75 (0.44)	0.31 (0.24)	v0.44 (0.36)	--
Special findings sub-score	0.06 (0.07)	0.06 (0.07)	0 (0.07)	--

The mean morphologic structural scores from the two independent readers are presented. Perfusion was not assessed. The morphology structure score ranges from 0–60, with higher values indicating more severe disease. Statistical analyses not performed on abscess/sacculation sub-score, consolidation sub-score, special findings sub-score due to the limited number of lobes assessed containing one of these findings. E/T/I: elexacaftor/tezacaftor/ivacaftor; SD: standard deviation.

**Table 3 jcm-11-06160-t003:** Structural Improvement by lobe of lung.

	Baseline (Mean, SD)	Post E/T/I (Mean, SD)	*p*-Value
Right upper lobe	2.06 (0.98)	1.38 (1.22)	0.2
Right middle lobe	1.44 (0.98)	0.75 (0.6)	0.008
Right lower lobe	0.31 (0.59)	0.06 0.18)	0.23
Left upper lobe	2 (2.1)	1.19 (1.65)	0.02
Lingula	0.5 (0.76)	0.75 (0.96)	0.275
Left lower lobe	0.5 (0.6)	0.13 (0.35)	0.048

Baseline structural scores by lobe, compared with scores at follow-up. E/T/I: elexacaftor/tezacaftor/ivacaftor; SD: standard deviation.

**Table 4 jcm-11-06160-t004:** Functional ventilation parameters before and after treatment.

	Baseline Mean (SD)	Post E/T/I Mean (SD)	Mean between Visit Change	*p*-Value
VDP	8.36% (4.81)	5.64% (3.53)	2.71% (2.11)	0.008
FLV↑tau2	5.04% (1.84)	3.38% (1.84)	1.66% (1.64)	0.024
Combined abnormal ventilation score	13.4% (6.93)	9.03% (5.19)	4.37% (2.78)	0.002

E/T/I: Elexacaftor/tezacaftor/ivacaftor. FLV↑tau2: fractional lung volume with prolonged gas washout time. SD: standard deviation. VDP: ventilation defect percentage.

**Table 5 jcm-11-06160-t005:** Correlations between imaging endpoints and lung function measurements (FEV_1_ and LCI) at each visit.

	FEV1 at Baseline	FEV1 Post E/T/I	Delta/Delta	LCI at Baseline	LCI Post E/T/I	Delta/Delta
Morphological structure score	−0.92 *	−0.90 *	0.17	0.81 *	0.80 *	0.27
Bronchiectasis/airway wall thickening score	−0.90 *	−0.91 *	0.24	0.77 *	0.84 *	−0.11
Mucus plugging score	−0.88 *	−0.26	−0.09	0.88 *	0.25	0.08
VDP	−0.86 *	−0.85 *	0.12	0.61	0.68	−0.22
FLV↑tau2	−0.81 *	−0.93 *	−0.58	0.66 ^+^	0.87 *	0.41
Abnormal Ventilation score	−0.89 *	−0.92 *	−0.27	0.66 ^+^	0.77 *	0.08

Correlations (r) between FEV_1_ and LCI at each visit with other study parameters at baseline and follow-up, as well as the change between visits. * indicates *p* < 0.05. ^+^ indicates *p* = 0.08. E/T/I: elexacaftor/tezacaftor/ivacaftor. FEV_1_: forced expiratory volume in 1 s. LCI: lung clearance index.

## Data Availability

Data are available upon request from the authors.

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
