# Peer review of "Dynamic Perfluorinated Gas MRI Shows Improved Lung Ventilation in People with Cystic Fibrosis after Elexacaftor/Tezacaftor/Ivacaftor: An Observational Study"

_jcm, 2022, doi:10.3390/jcm11206160_

Round 1

Reviewer 1 Report

The whole document has a lot of abbreviations that it is not sometimes necessary. You only include abbreviations in the most used words to be easier reading. 

The lung disease under study, cystic fibrosis, must appear in the title. In addition, the title sounds like a clinical trial, you should change it. 

Line 67: H UTE is not described until line 87 such as H ultrashort echo time (UTE). In line 67 you must explain it, and the abbreviation must be the same throughout the text. 

Line 142-145: The authors must include a flow diagram of participants during the study.

Is baseline the visit 1 (before starting E/T/I)?   And is the post E/T/I the visit 2 after a month? You should clarify the times, for example, using the same terms during the study.

The tables need a table legend with the meaning of the included abbreviations. Table 4 doesn’t include the value data, is mean (SD)?

The conclusion is not in line with the objective of the study. 

Reviewer 2 Report

This is an interesting pilot study describing the effect of Elexacaftor/Tezacaftor/Ivacaftor on structural and functional changes in MRI.

Although the subject is interesting, there are several major concerns that preclude the acceptance of the manuscript in its current form.

1.       The study group is very small. Although the results are interesting, is should be stated more clearly that the small number is a main limitation.

2.       Although the use of MRI has some advantages, the gold standard for evaluating lung tissue has been, and still is chest CT. This subject merits further discussion. What are the advantages and disadvantages of MRI compared to CT?  Did any of the patients have CT scoring before the initiation of Trikafta?

For example - papers I would add to such a discussion are PMID: 24065629 and PMID: 16941092.

3.       The group of patients in the study were adults with mild lung disease. Thus, the results should be interpreted with caution. It is not clear if the results can be generalized to pediatric patients, and to patients with more severe lung disease. Can more severe bronchiectasis be reversible in such a short period of time? Are there any studies evaluation the time frame of structural changes after the initiation of mutation-specific therapy?

4.       It would be Interesting to repeat the examinations after a few months of therapy – to evaluate the structural and functional changes over time.
